# Direct asymmetric α C($sp^3$)–H alkylation of benzylamines with MBH acetates enabled by bifunctional pyridoxal catalysts

Jiayao Chen[1,3], Yue Yang[1,2,3], Siqi Liu [1], Weibo Ling[1], Ruixin Zhang[1], Tongyin Chen[1], Wen-Wen Chen [1] ✉ & Baoguo Zhao [1,2] ✉

Organocatalytic allylic substitution of Morita-Baylis-Hillman (MBH) adducts is widely regarded as one of the most powerful transformations in organic synthesis. A range of activated carbon nucleophiles have been successfully employed in this reaction, enabling the incorporation of diverse functional moieties. Despite its potential, the use of inert C–H nucleophiles—critical for broadening the reaction's versatility and synthetic utility—remains a significant challenge in the field. Direct α-C–H functionalization of benzyl amines with MBH adducts offers a promising route to form a new C–C bond while simultaneously establishing a chiral amine moiety, a feature highly attractive from the perspective of organic synthesis. However, this transformation is particularly challenging due to the inherent inertness of the α-C($sp^3$)–H bonds, significant nucleophilic interference from the NH₂ group, and the complexity of selectivity control. Herein, we have successfully achieved an asymmetric direct α-C–H allylic alkylation of NH₂-unprotected benzylamines with MBH adducts using a bifunctional chiral pyridoxal catalyst, producing biologically important chiral γ-amino acid derivatives in good yields with excellent diastereo- and enantioselectivities. The reaction offers a distinct strategy for synthesizing multiply functionalized compounds from readily available starting materials, significantly expanding access to complex chiral architectures.

Organocatalytic allylic substitution of Morita-Baylis-Hillman (MBH) adducts has emerged as a highly versatile synthetic transformation[1–7], allowing the introduction of a wide diversity of nucleophiles into multifunctional molecular frameworks. While activated carbon-nucleophiles have been widely utilized[8–22], the use of inert C–H nucleophiles, essential for expanding the reaction's versatility and synthetic utility, remains a major challenge[23]. Benzylamines are a class of readily accessible primary amines possessing two inert α C($sp^3$)–H bonds[24]. Direct asymmetric organocatalytic α-C($sp^3$)–H allylic alkylation of benzylamines with MBH adducts offers a promising strategy for simultaneous formation of a C–C

bond and establishment of chiral amine functionality (Fig. 1a), providing an efficient and appealing approach to construct chiral substituted γ-aminobutyric acid (GABA) scaffolds that are prevailing core structures of numerous bioactive natural products and pharmaceutical molecules (Fig. 1b)[25–29]. Moreover, through simple cyclization, γ-aminobutyric acids can be easily converted into another type of biologically important γ-lactam analogs (Fig. 1b)[30–32]. In spite of its potential benefits, this transformation remains undeveloped and is a significant challenge in organic chemistry, even when employing NH₂-protected benzylamine derivatives. The difficulties can be attributed to the following two factors

[1]The Education Ministry Key Lab of Resource Chemistry, Shanghai Frontiers Science Center of Biomimetic Catalysis and College of Chemistry and Materials Science, Shanghai Normal University, Shanghai, China. [2]Frontiers Science Center for Transformative Molecules and The School of Chemistry and Chemical Engineering, Shanghai Jiao Tong University, Shanghai, China. [3]These authors contributed equally: Jiayao Chen, Yue Yang. ✉e-mail: wenwen@shnu.edu.cn; zhaobg2006@shnu.edu.cn

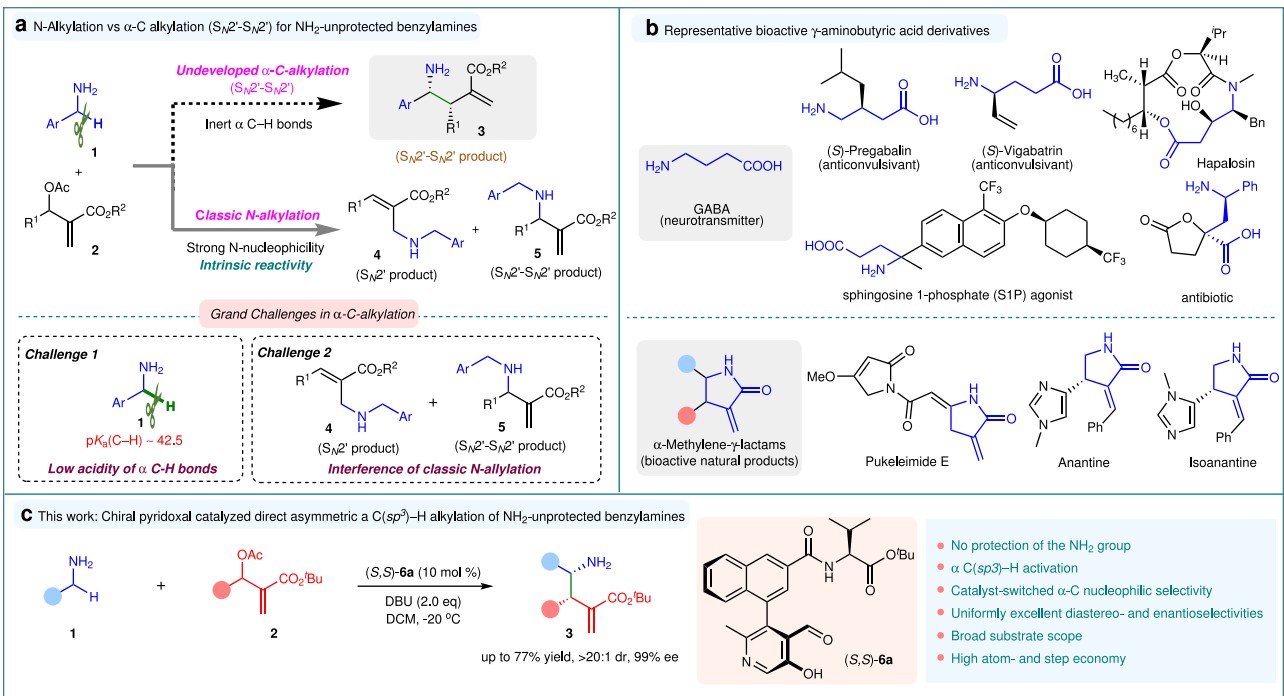

**Fig. 1 | α C–H allylic alkylation of benzylamines with MBH adducts.**
**a** N-Alkylation vs α-C alkylation ($S_N2'$-$S_N2'$) of NH$_2$-unprotected benzylamines.
**b** Representative bioactive γ-aminobutyric acid derivatives. **c** This work: chiral pyridoxal catalyzed direct asymmetric α C($sp^3$)–H alkylation of NH$_2$-unprotected benzylamines with MBH acetates. Ar, aromatic group; Ph, phenyl; $S_N2'$, bimolecular allylic nucleophilic substitution; DBU, 1,8-diazabicyclo[5.4.0]undec-7-ene; DCM, dichloromethane; dr, diastereomeric ratio; ee, enantiomeric excess.

(Fig. 1a). First, deprotonating the α-amino C–H bonds to produce active carbanions for initiating the addition is rather difficult due to the extremely low acidity of α-amino C–H bonds ($pK_a \sim 42.5$)[24]. Second, owing to the high nucleophilicity of the NH$_2$ group, the classical N-substitution[33,34] usually disrupts the desired α-C alkylation, resulting in the predominant formation of the N-alkylated products **4** and/or **5** (Fig. 1a).

Carbonyl catalysis[35–40] has become an effective strategy for direct α C–H functionalization of NH$_2$-unprotected primary amines[41–50]. With chiral pyridoxals as the carbonyl catalysts, a series of enantioselective transformations of activated primary amines, such as α-amino esters with diverse electrophiles, have been successfully accomplished, affording diverse chiral amino acid derivatives[41–43]. Nevertheless, direct asymmetric α-functionalization of primary amines containing inert α-amino C–H bonds[24,44,45], for instance, benzylamines, presents greater difficulties. Till now, for the carbonyl-catalyzed reaction of benzylamines, only the asymmetric addition to aldehydes has been achieved[24]. Allylic alkylation of benzylamines with MBH acetates is more difficult to achieve, primarily due to the strong interference from the highly nucleophilic amino group and the more complicated functional group compatibility of MBH acetates. Herein, we would like to disclose our success on the direct asymmetric α C–H alkylation of benzylamines **1** with MBH acetates **2**, with a switched nucleophilic selectivity from N to α-C of benzylamines **1**, enabled by the chiral bifunctional pyridoxal catalysts **6**[41] bearing an amide side chain attached to the C3 position of the naphthyl ring. This reaction produced a wide range of chiral polysubstituted γ-amino acid esters **3** with excellent diastereo- and enantioselectivities (up to >20:1 dr, 99% ee) (Fig. 1c).

## Results

### Reaction optimization
Our studies commenced with the investigation of the direct asymmetric α C–H alkylation of benzylamine (**1a**) with MBH acetate **2a**

(Fig. 2, Supplementary Table S1). To our delight, with pyridoxal (*S*,*S*)-**6a** as the catalyst and DBU as the base, the reaction proceeded smoothly for 24 h as anticipated to afford chiral γ-amino acid ester **3a** in a 58% yield with >20:1 diastereomeric ratio (dr) and 99% ee for the major diastereomer accompanied by the formation of some N-alkylated by-products (Fig. 2, Supplementary Table S1, entry 1). The pyridoxal catalyst is crucial for this reaction, as no desired α-C allylic alkylation but only N-alkylation can be observed in its absence (Supplementary Table S1, entries 2 and 3), demonstrating that the pyridoxal catalyst is capable of switching the nucleophilic selectivity of benzylamine **1a** from N to α-C without protecting the NH$_2$ group. Extending the reaction time to 72 h can effectively improve the yield of the reaction to 71% while maintaining the excellent diastereo- and enantioselectivities (> 20:1 dr, 99% ee) (Fig. 2, Supplementary Table S1, entry 4). The diastereomeric pyridoxal (*R*,*S*)-**6a** was less effective for the reaction, resulting in the product **3a** with decreased yield and stereoselectivity (Fig. 2). Among the pyridoxals **6a-e** examined, compound (*S*,*S*)-**6a** displayed the best performance regarding activity, diastereo-, and enantioselectivities (Fig. 2). Pyridoxals (*R*,*S*)-**7** and (*S*,*S*)-**7** possessing a lateral amide chain at the C2 position of the naphthyl ring were completely ineffective for the reaction (Fig. 2), indicating the side chain is important for the activity of the pyridoxal catalyst. Reaction condition investigations exhibited that DBU was the base of choice (Supplementary Table S1, entry 1 vs 12-16) and dichloromethane was the optimal solvent (Supplementary Table S1, entry 1 vs 17-20).

### Substrate scope
Under the optimal reaction conditions, substrate scope on benzylamines was investigated (Fig. 3). A variety of benzylamines **1** bearing different electron-donating and/or electron-withdrawing substituents at the *ortho*-, *meta*-, or/and *para*-positions of the benzene ring all smoothly underwent the direct asymmetric α C–H allylic alkylation with MBH acetate **2a**, producing the corresponding products **3b-m** in

**Fig. 2 | Catalyst screening.** Reaction conditions: **1a** (0.20 mmol), **2a** (0.10 mmol), catalyst **6** (0.01 mmol, 10 mol%), and DBU (0.20 mmol) in DCM (0.5 mL) at -20 °C for 72 h. Isolated yields were based on **2a**. The dr values were determined by $^1$H NMR analysis of the crude reaction mixtures. The ee values were determined by chiral HPLC analysis. $^i$Pr: isopropyl; $^t$Bu: *tert*-butyl; Ph: phenyl; Bn: benzyl; Ac, acetyl; DBU, 1,8-diazabicyclo[5.4.0]undec-7-ene; DCM, dichloromethane. $^a$The reaction time was 24 h.

good yields (60-77%) with excellent diastereo- and enantioselectivities (> 20:1 dr and 96-99% ee) (Fig. 3a). The absolute configuration of the major diastereomer of product **3m** was determined as (3*S*,4 *R*) by X-ray analysis (Fig. 3a). Delightfully, the reaction is not sensitive to the steric hindrance. The *ortho*-substituted benzylamine, such as *o*-tolylmethanamine (as for **3j**), still displayed good reactivity with high stereoselectivity (> 20:1 dr, 98% ee). Naphthyl substituted methanamines (as for **3n** and **3o**) as well as heteroarylmethanamines such as thiophen-3-ylmethanamine (as for **3p**), pyridin-3-ylmethanamine (as for **3q**), and (4-methoxypyridin-2-yl)methanamine (as for **3r**) were also applicable for the reaction. The other reaction partner MBH acetates were also examined. Phenyl (as for **3s**), substituted phenyl (as for **3t**–**ad**) and heteroaromatic (as for **3ae**–**ah**) MBH acetates with different electronic properties and substitution patterns on the benzene ring were proven to be effective substrates for the α-C allylic substitution, as they all exhibited good reactivities and led to the formation of products **3s**–**ah** with uniformly high diastereomeric ratios (> 20:1 dr) and high enantiopurities (94-98% ee) (Fig. 3b). The electronic property of the substituted phenyl groups seems to have little impact on the diastereo- and enantioselectivity. Alkenyl- and alkynyl-substituted MBH acetates are also reactive for the transformation, producing chiral γ-amino acid esters **3ai** and **3aj** with excellent diastereoselectivity (Fig. 3b). The relatively low ee value observed for **3aj** is likely due to the linear geometry of the alkyne group, which lacks sufficient steric hindrance for effective enantiocontrol. Alkyl-substituted MBH acetates, such as *tert*-butyl 3-acetoxy-2-methylene-5-phenylpentanoate, are completely ineffective for the α-C allylic alkylation. Notably, when utilizing substrates containing a biologically active chiral moiety derived from D-glucose (as for **3ak**), testosterone[51] (as for **3al**), or estrone (as for **3am**), the reaction proceeded fluently to afford products **3ak-am** in good yields with excellent diastereocontrol (Fig. 3c).

## Synthetic utility

To demonstrate the practical utility of the protocol, the reaction was carried out on a gram-scale. Chiral γ-amino acid ester **3h** (1.051 g) was obtained in a comparable yield with the same diastereo- and enantioselectives (Fig. 4a). The synthetic utility of the product was further investigated. Chiral γ-amino acid esters **3** can be facilely transformed into different derivatives with potential bioactivities (Fig. 4b). Deprotection of the *tert*-butyl moiety of the major diastereomer (3*S*,4*R*)-**3h** by means of HCl produced chiral γ-aminobutyric acid **8h** in 95% yield while maintaining the same enantiopurity. As illustrated in Fig. 4b, cyclic α-methylene-γ-lactam **9a** and **9h** were successfully obtained in satisfactory yields with high enantiopurities via sequential deprotection and condensation. Notably, α-methylene-γ-lactam derivatives have been discovered to exhibit anti-inflammatory, phytotoxic, cytotoxic, and antimicrobial bioactivities[30–32]. Additionally, under Pd-catalyzed hydrogenation conditions, the reduction of **3h** proceeded smoothly to afford a pair of chromatographically separable diastereomers (2*R*,3*R*,4*R*)-**10h** (46% yield) and (2*S*,3*R*,4*R*)-**10h'** (32% yield) without any loss of enantiopurity, as depicted in Fig. 4b. Furthermore, compound **10h'** can be converted into cyclic γ-lactam **11h'** as presented in Fig. 4b. The absolute configuration of compounds **11h'** as well as **9a** were confirmed by X-ray analysis (see Supplementary Information).

## Mechanistic studies

The reaction was proposed to proceed through a carbonyl catalysis mechanistic pathway, in which the chiral pyridoxal catalyst served as a pivotal factor (Fig. 5a)[35–39]. The condensation between chiral pyridoxal catalyst **6a** and arylmethanamine **1** results in the formation of imine **12**, which activates the benzylic C−H bonds and remarkably increases the C−H acidity for further deprotonation to generate delocalized

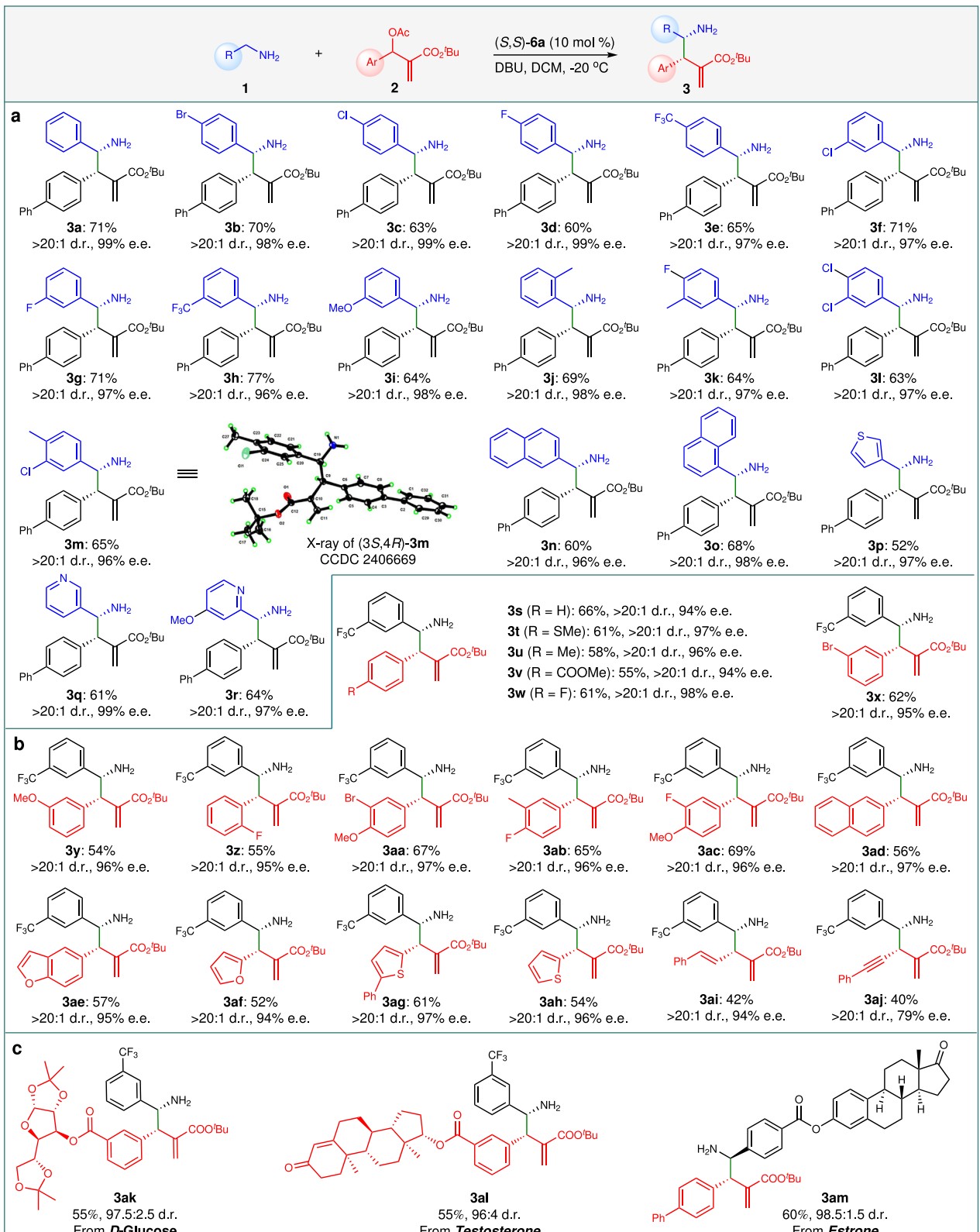

**Fig. 3 | Substrate scope. a** Investigation on benzylamines. **b** Investigation on MBH acetates. **c** Investigation on chiral substrates. Reaction conditions: **1** (0.40 mmol), **2** (0.20 mmol), (*S*,*S*)-**6a** (0.02 mmol, 10 mol%) and DBU (0.40 mmol) in DCM (1.0 mL) at −20 °C for 72 h. The isolated yields were based on **2**. The dr values were determined by ¹H NMR analysis of crude reaction mixtures. The ee values were determined by chiral HPLC analysis. The absolute configuration for **3m** was determined as (3*S*,4*R*) by X-ray analysis, and those for **3a**-**3l**, **3n**-**3am** were tentatively assigned by analog. DBU, 1,8-diazabicyclo[5.4.0]undec-7-ene; DCM, dichloromethane.

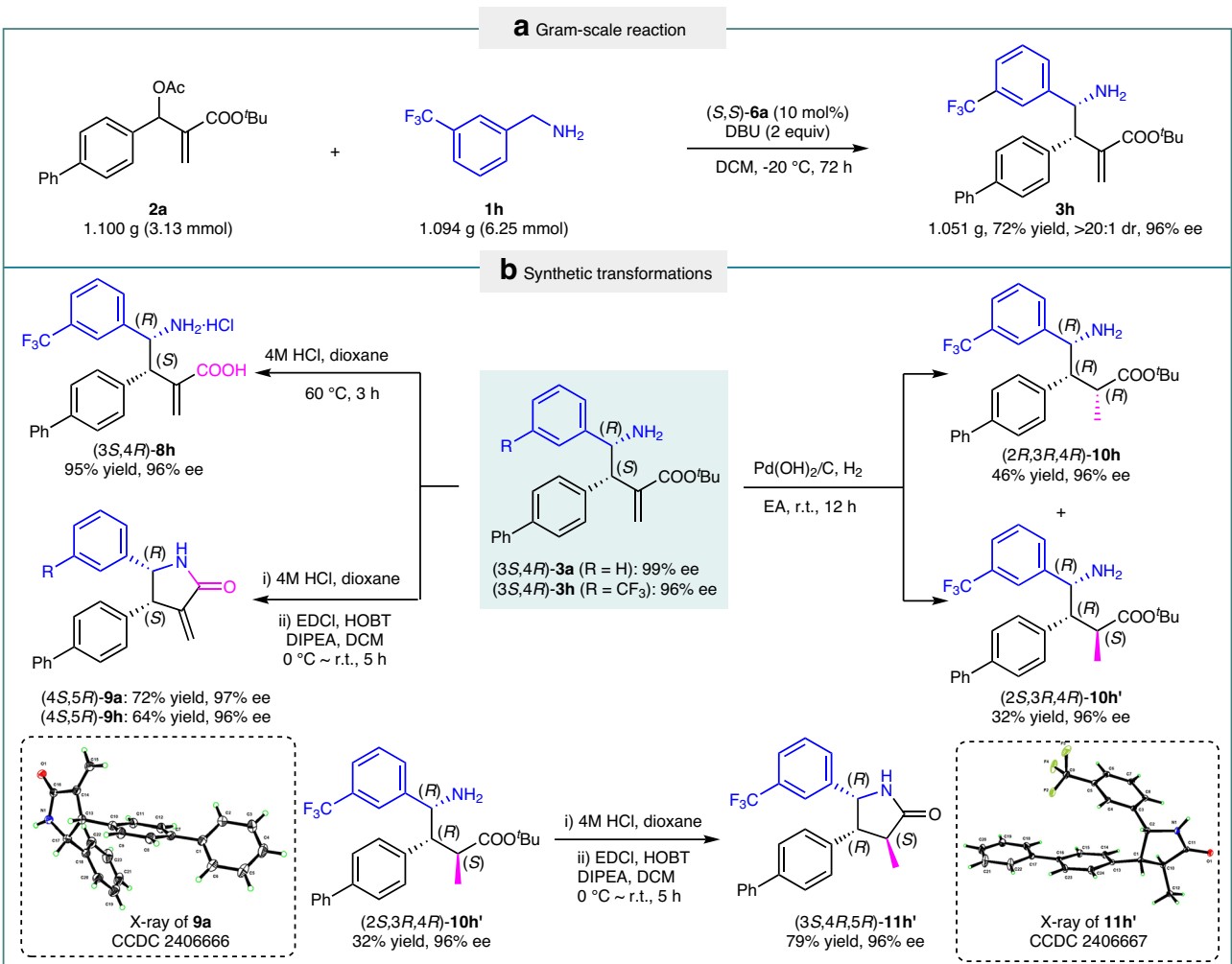

**Fig. 4 | Synthetic applications. a** Gram-scale reaction. **b** Synthetic transformations. Ac, acetyl; EDCl, 1-(3-dimethylaminopropyl)-3-ethylcarbodiimide hydrochloride; HOBT, 1-hydroxybenzotriazole; DIPEA, N, N-Diisopropylethylamine; EA, ethyl acetate; DBU, 1,8-diazabicyclo[5.4.0]undec-7-ene; DCM, dichloromethane.

carbanion 13[52,53]. Computational p$K_a$ analysis reveals the benzylic C–H of imine 12 is more acidic than the phenolic O–H and N–H of the pyridoxal scaffold, thermodynamically favoring the formation of the carbanion via selective C–H deprotonation (see Supplementary Information for details). The carbanion 13 undergoes an asymmetric addition reaction to quaternary ammonium salt intermediate 14, which is generated via the nucleophilic addition of DBU to MBH acetate 2. The addition is accompanied by the expulsion of DBU, leading to the formation of species 16. Hydrolysis of compound 16 leads to the liberation of the $S_N2'$-$S_N2'$ α-C alkylation product 3 and the regeneration of the pyridoxal catalyst 6a, completing the catalytic cycle.

Kinetic isotope effect (KIE) studies were conducted with equimolar amounts of benzylamine 1a and α-deuterated counterpart 1a-*d* (Fig. 5b). The reaction led to the formation of compound 3a and the deuterated 3a-*d*, with a ratio of 7.69:1. The distinct kinetic isotope effect strongly suggests that the deprotonation of imine 12 to afford the active carbanion intermediate 13 is the rate-determining step for the entire reaction pathway.

To elucidate the origin of chirality, computational investigations have been conducted. Figure 5c depicts the optimized transition state (15) for the step involving the asymmetric addition of carbanion 13 to intermediate 14. While benzylamine 1 is bound by the catalyst via the formation of the imine with the pyridoxal moiety, the MBH acetate-derived intermediate 14 is activated by the side chain of the pyridoxal catalyst. This activation occurs via a hydrogen-bonding interaction[54-56] between the NH group of the side chain and the carbonyl group of intermediate 14. To minimize steric repulsion, intermediate 14 adopts an orientation wherein both the bulky R group and the DBU moiety are directed away from the biaryl backbone of the pyridoxal catalyst. The carbanion originating from arylmethanamines approaches 14 from above to afford chiral γ-amino acid esters 3 with (3S,4R)-configuration from chiral pyridoxal (S,S)-6a.

The postulated transition state is supported by control experiments on the comparison of catalysts (Fig. 5d). Methylation of the amide N–H group on the lateral side chain of pyridoxal 6a resulted in a significant decline in activity and enantioselectivity. As shown in Fig. 5d, the reaction with catalyst 6f afforded a 10% yield, a diastereomeric ratio (dr) of >20:1, and an enantiomeric excess (ee) of 53%, in contrast to the 34% yield, >20:1 dr, and 92% ee obtained with catalyst 6e. The result implies that the amide N–H group likely participates in the catalytic process via hydrogen bonding, as proposed in transition state 15. The cooperative bifunctional activation accounts for the excellent performance of pyridoxals 6 bearing a C3 amide side chain in the reaction. This was further confirmed by the fact that pyridoxals 7, having a C2 amide side chain, are completely inactive for the reaction (Fig. 2). It is supposed that the side chain is too close to the aldehyde moiety to guarantee effective bifunctional activation during the catalysis.

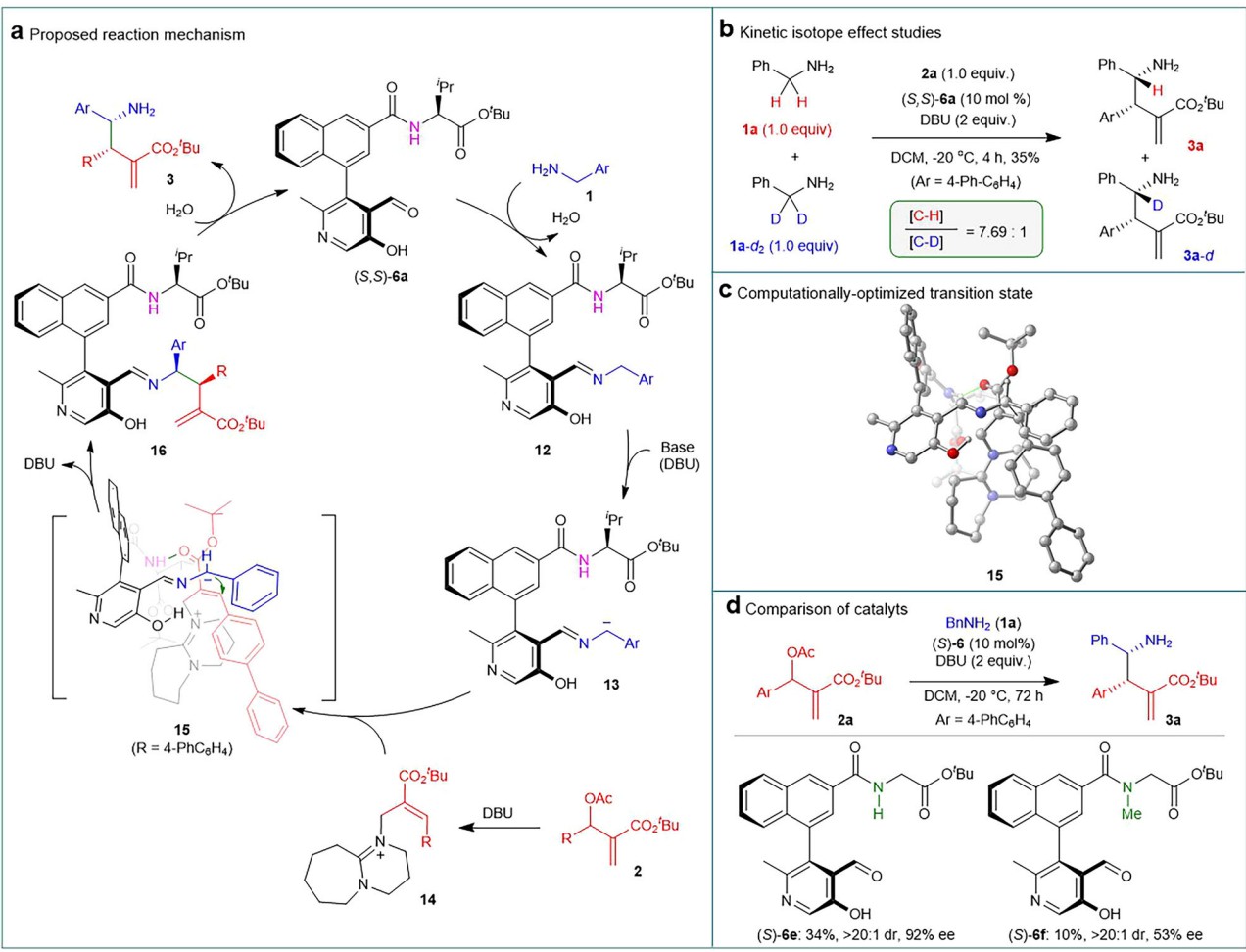

**Fig. 5 | Mechanistic studies. a** Proposed reaction mechanism. **b** Kinetic isotope effect studies. **c** Computationally-optimized transition state. **d** Comparison of catalysts. DBU, 1,8-diazabicyclo[5.4.0]undec-7-ene; DCM, dichloromethane.

## Discussion

In summary, we have successfully developed a direct asymmetric α C−H allylic alkylation of benzylamines **1** with MBH acetates **2** by utilizing bifunctional chiral pyridoxal (S,S)-**6a** as the catalyst. This reaction furnishes diverse chiral γ-amino acid esters **3**, which are of biological significance, in 40–77% yields with excellent diastereo- (> 20:1 dr) and enantioselectivities (79–99% ee). This work exemplifies a remarkable illustration of organocatalyzed inert C−H bond functionalization under mild conditions, offering high stereocontrol without directing or protecting group manipulations. Moreover, it also highlights the distinctive exceptional capabilities of vitamin B6[57,58] based organocatalysts[59–61] in organic synthesis.

## Methods

### General procedure for asymmetric α C(sp³)–H allylic alkylation of benzylamines with MBH acetates (Fig. 3)

To a 4 mL vial equipped with a magnetic stirrer bar were successively added chiral pyridoxal (S,S)-**6a** (0.0046 g, 0.010 mmol), DBU (0.0304 g, 0.20 mmol), DCM (0.3 mL) and benzylamine **1** (0.20 mmol). The mixture was stirred at −20 °C for 5 min, and a solution of MBH acetate **2** (0.10 mmol) in DCM (0.2 mL) was added in portions over 1 h. After the reaction mixture was stirred at −20 °C for 72 h, it was allowed to warm up to room temperature and concentrated via rotary evaporator to remove most of the solvent. Then it was dried under vacuum and submitted to ¹H NMR analysis to determine the dr values. The product **3** was purified by column chromatography on silica gel (petroleum ether: ethyl acetate = 3:1). The dr values of products **3a-ak** were determined by ¹H NMR analysis of the crude reaction mixtures. The enantiomeric excesses (ee's) of products **3a-ak** were determined by chiral HPLC analysis.

### Procedure for synthesis of γ-amino acid ester 3 h in gram-scale (Fig. 3a)

To a 25 mL flask equipped with a magnetic stirrer bar were successively added chiral pyridoxal (S,S)-**6a** (0.144 g, 0.312 mmol), DBU (0.950 g, 6.250 mmol), DCM (9.35 mL) and arylmethanamine **1h** (1.094 g, 6.250 mmol). The mixture was stirred at −20 °C for 5 min, and a solution of MBH acetate **2a** (1.100 g, 3.130 mmol) in DCM (6.25 mL) was added in portions over 1 h. After the reaction mixture was stirred at −20 °C for 72 h, it was allowed to warm up to room temperature and concentrated via rotary evaporator to remove most of the solvent. Then it was dried under vacuum and submitted to ¹H NMR analysis to determine the dr values. The crude reaction mixture was purified by column chromatography on silica gel (petroleum ether: ethyl acetate = 3:1) to afford compound (S,R)-**3h** (1.051 g, 72% yield, >20:1 dr, 96% ee) as a white solid.

## Data availability

The authors declare that the data supporting the findings of this study are available within the article and Supplementary Information file, or from the corresponding author upon request. The X-ray crystallographic coordinates for structures reported in this study have been deposited at the Cambridge Crystallographic Data Center (CCDC), under deposition numbers of CCDC 2406669 [(3S,4R)-**3m** in

Supplementary Fig. S8], CCDC 2406666 [(4*S*,5*R*)-**9a** in Supplementary Fig. S9] and CCDC 2406667 [(3*S*,4*R*,5*R*)-**11h'** in Supplementary Fig. S10]. These data can be obtained free of charge from The Cambridge Crystallographic Data Center via https://www.ccdc.cam.ac.uk/structures/. Coordinates of the optimized structures are available from the Supplementary Data 1.

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

## Acknowledgments

We are grateful for the generous financial support from the National Key R&D Program of China (2023YFA1506402, B.Z.), National Natural Science Foundation of China (NSFC) (22271192, B.Z.; 22271191, W.C.), the Shanghai Municipal Science and Technology Major Project, the Shanghai Municipal Committee of Science and Technology (23ZR1446300, W.C.; 24ZR1456200, S.L.), and the Shanghai Engineering Research Center of Green Energy Chemical Engineering (18DZ2254200, B.Z.).

## Author contributions

B.Z. conceived and directed the project and wrote the paper. W.C. co-directed the project and co-wrote the manuscript. J.C. and Y.Y. performed the experiments and most of the Supplementary Documents. S.L. and T.C. conducted the DFT calculations. W.L. and R.Z. prepared part of the Supplementary Documents.

## Competing interests

The authors declare no competing interests.
