## [Transparent Peer Review file · Nature Communications]

Direct Asymmetric α C(sp³)-H Alkylation of Benzylamines with MBH Acetates Enabled by Bifunctional Pyridoxal Catalysts

Corresponding Author: Professor Baoguo Zhao

Version 0:

Reviewer comments:

Reviewer #1

(Remarks to the Author)

This manuscript by Zhao and Chen describes an enantioselective α C(sp³)-H alkylation of NH₂-unprotected benzylamines with MBH acetates via carbonyl catalysis, yielding a wide range of biologically important chiral γ -amino acid derivatives with good yields and excellent diastereo- and enantioselectivities. By employing a biomimetic chiral bifunctional pyridoxal as the carbonyl catalyst, the reaction has successfully overcome the inert reactivity of the α -C-H bonds of the benzylamines, as well as the nucleophilic interference from the active NH₂ group. This has led to the development of an intriguing and novel reactivity between primary benzylamines and MBH acetates, significantly enhancing the applicability of allylic substitution reactions of MBH adducts with inert nucleophiles. It is highly impressive that this work utilizes an organocatalyst to achieve direct activation of inert α -amino C-H bonds without the assistance of a transition-metal catalyst, expanding the scope of organocatalysis in chiral synthesis. Therefore, I would like to recommend acceptance of this manuscript to publish in Nature Communications, after addressing the following concerns and questions.

1. In this manuscript, aromatic and heteroaromatic MBH acetates are well tolerated in the reaction. The authors are encouraged to examine more substrates such as alkenyl or alkynyl substituted MBH acetates. If alkyl substituted MBH acetates can be applicable in the reaction, it will highly broaden the substrate scope to a greater extent.
2. The NMR spectra of 3j and 3p in SI show signals that hint to some impurities. Please re-purify those compounds to give the pure spectrum.
3. There are some errors and typos need to be corrected.
 - 1) Page 1, line 23, "unprotected NH₂-benzylamines" should be "NH₂-unprotected benzylamines".
 - 2) Page 9, line 142, "Synthetic transformations" should be "Synthetic applications". The caption should be in accordance with the figure.
 - 3) Page 11, line 170, "beneath" should be "above".

Reviewer #2

(Remarks to the Author)

In this paper, authors have developed an asymmetric direct α -C-H allylic alkylation of unprotected NH₂-benzylamines with MBH adducts using a bifunctional chiral pyridoxal catalyst to obtain chiral γ -amino acid derivatives in good yields with excellent stereoselectivities. Notably, this transformation is particularly challenging due to the inherent inertness of the α -C(sp³)-H bonds, significant nucleophilic interference from the NH₂ group, and the complexity of selectivity control. In addition, both the practical utility and reaction mechanism of the protocol was also investigated. The manuscript and SI were well prepared. Overall, it was suggested to be published in nature communication after the formatting the reference.

Reviewer #3

(Remarks to the Author)

In this manuscript, Zhao and co-workers report the development of an enantio- and diastereoselective α -C–H allylic alkylation of benzylamines using a bifunctional chiral pyridoxal catalyst derived from vitamin B6. Building on previous work by Jianfeng Chen and colleagues, the authors propose that the asymmetric C(sp³)–H alkylation with MBH acetates proceeds via a carbonyl-catalysis activation mode. Although the manuscript is well written, and the methodology presented is synthetically valuable—highlighting the potential of bioinspired catalysts for the functionalization of otherwise inert C–H bonds—several mechanistic issues remain to be addressed. Therefore, I believe the work possesses the merits and originality necessary for publication in Nature Communications. However, I recommend reconsideration after major revisions that specifically address the mechanistic and methodological concerns outlined below. In this context, I will restrict my comments to the computational part of this work.

Remark 1: The authors propose a stereodetermining TS structure (TS15, Fig. 5c) to rationalize the high diastereo- and enantioselectivity observed in the experimental study, with a computed $\Delta\Delta G^\ddagger > 6.0$ kcal/mol between the competing TS structures (Fig. S1, SI). While these results are in good agreement with the experimental stereochemical outcome, the origin of this significant energetic difference must be analyzed in more detail. In particular, the authors do not evaluate whether the observed selectivity is driven by favorable non-covalent interactions (e.g., hydrogen bonding, dispersion, CH– π interactions) in the favored pathway (TS-S,R) or by destabilizing repulsions in the disfavored one. Therefore, the authors are encouraged to provide mechanistic insight into the observed selectivity using quantitative tools such as energy decomposition analysis (e.g., EDA-NOCV or NBO analysis), together with visualization tools such as NCIPLOT or van der Waals contact mapping. Such analyses would offer deeper mechanistic insight into the stereochemical control exerted by the chiral catalyst.

Remark 2: Related to the above, it appears that the authors did not conduct a thorough exploration of the conformational space for the reactive intermediate (structure 13 in Fig. 5a) and the TS structures. This may have led to a bias in selecting conformations that best support the proposed mechanism. Consequently, the reported $\Delta\Delta G^\ddagger$ values may be overestimated due to the omission of low-lying competing TS structures. Therefore, the authors are strongly encouraged to perform a more comprehensive conformational search for all relevant species to ensure the reliability of the computed selectivity.

Remark 3: A notable experimental observation is the exclusive formation of the α -C–H alkylation product, with no evidence of N-alkylation of the benzylamine via a direct SN2' pathway. However, the authors do not address the observed chemoselectivity through DFT calculations. I recommend including a computational comparison between the imine-formation pathway and a plausible SN2' attack of the free benzylamine on the MBH acetate-derived intermediate. Addressing this aspect is crucial to provide mechanistic insights that could guide related transformations.

Remark 4: The proposed mechanism states that, upon imine formation between the catalyst and the benzylamine, the acidic C–H bond at the benzylic position is selectively deprotonated to generate the reactive carbanion in situ. However, the authors do not provide a rationale to explain why this deprotonation is favored over alternative acidic sites, particularly the phenolic OH group of the pyridoxal scaffold. A computational pK_a comparison or an energy profile for the competitive deprotonation pathways would strengthen the mechanistic proposal.

Remark 5: As a final remark, I strongly recommend reoptimizing the key stereodetermining TS structures using a range-separated hybrid functional such as ω B97X-D, which has shown excellent performance in capturing dispersion and electrostatic interactions relevant in organocatalysis. Moreover, I recommend that the Gibbs free energies discussed in the manuscript and Supporting Information be corrected using the quasi-rigid rotor harmonic oscillator (QRRHO) approach, as proposed by Grimme, to provide more accurate thermodynamic values.

Version 1:

Reviewer comments:

Reviewer #1

(Remarks to the Author)

The authors had addressed issues suggested by the reviewers. The revised manuscript looks fine. The reviewer recommends accepting the manuscript for publication in this journal.

Reviewer #3

(Remarks to the Author)

The authors have carefully and satisfactorily addressed all the concerns raised in the initial round of review. The revised version provides a significantly improved mechanistic description, supported by new DFT calculations. Therefore, I find the current version of the manuscript suitable for publication in Nature Communications. However, I would like to raise two minor points that the authors may consider prior to publication.

(1) In response to Remark 3, the authors computed the transition state for the addition of benzylamine to the MBH adduct, reporting an activation free energy (ΔG^\ddagger) of 20.8 kcal/mol. While this information is valuable, the manuscript would benefit from a comparison with the barrier for the addition of benzylamine to the pyridoxal derivative (S,S)-6a, leading to intermediate 12, as previously suggested by this reviewer. Such a comparison would provide a more complete view of the chemoselectivity in the early stage of the mechanism.

(2) Regarding the EDA analysis, the authors identify that the orbital interaction term (ΔE_{orb}) is significantly more stabilizing

in TS1-SR compared to TS1-RS. At first sight, this difference could be ascribed to a shorter forming C–C bond in TS1-SR. In this context, it would be informative to explore whether noncovalent interactions or steric effects are responsible for limiting a more favorable nucleophilic approach in TS1-RS. A qualitative analysis using NCI-plot could help clarify whether the spatial arrangement of aryl groups in TS1-RS hinders optimal orbital overlap through repulsive interactions.

Version 2:

Reviewer comments:

Reviewer #3

(Remarks to the Author)

The authors have revised the manuscript and addressed all my concerns. The manuscript may be accepted for publication.

POINT-TO-POINT RESPONSE:

Response to Reviewer #1: Great thanks for the helpful comments and suggestions

(1) *In this manuscript, aromatic and heteroaromatic MBH acetates are well tolerated in the reaction. The authors are encouraged to examine more substrates such as alkenyl or alkynyl substituted MBH acetates. If alkyl substituted MBH acetates can be applicable in the reaction, it will highly broaden the substrate scope to a greater extent.*

Response: According to this suggestion, we have examined alkenyl-, alkynyl- and alkyl-substituted MBH acetates, including **2ai**, **2aj** and **2an**. Alkenyl and alkynyl substituted MBH acetates **2ai** and **2aj** respectively gave the corresponding chiral γ -amino acid esters **3ai** and **3aj** under the standard reaction conditions. The relatively low ee value observed for **3aj** is likely due to the linear geometry of the alkyne group, which lacks sufficient steric hindrance for effective enantiocontrol. Alkyl-substituted MBH acetate **2an** is an ineffective substrate for the reaction. The results and the corresponding discussions have been included in the main text and the Supporting Information.

(2) *The NMR spectra of 3j and 3p in SI show signals that hint to some impurities. Please re-purify those compounds to give the pure spectrum.*

Response: According to the suggestion, we have repurified the products **3j** and **3p**. The ¹H NMR and ¹³C NMR spectra of compounds **3j** and **3p** have been updated in SI.

(3) *There are some errors and typos need to be corrected.*

1) *Page 1, line 23, “unprotected NH₂-benzylamines” should be “NH₂-unprotected benzylamines”.*

2) *Page 9, line 142, “Synthetic transformations” should be “Synthetic applications”. The caption should be in accordance with the figure.*

3) *Page 11, line 170, “beneath” should be “above”.*

Response: According to the suggestions, we have checked the manuscript carefully and corrected the identified typo errors.

Response to Reviewer #2: Great thanks for the helpful comments and suggestions

(1) *Overall, it was suggested to be published in nature communication after the formatting the reference.*

Response: As suggested, we have formatted the reference according to the Nature Communications formatting instructions.

Response to Reviewer #3: Great thanks for the helpful comments and suggestions

(1) *Remark 1: The authors propose a stereodetermining TS structure (TS15, Fig. 5c) to rationalize the high diastereo- and enantioselectivity observed in the experimental study, with a computed $\Delta\Delta G^\ddagger > 6.0$ kcal/mol between the competing TS structures (Fig. S1, SI). While these results are in good agreement with the experimental stereochemical outcome, the origin of this significant energetic difference must be analyzed in more detail. In particular, the authors do not evaluate whether the observed selectivity is driven by favorable non-covalent interactions (e.g., hydrogen bonding, dispersion, CH- π interactions) in the favored pathway (TS-S,R) or by destabilizing repulsions in the disfavored one. Therefore, the authors are encouraged to provide mechanistic insight into the observed selectivity using quantitative tools such as energy*

decomposition analysis (e.g., EDA-NOCV or NBO analysis), together with visualization tools such as NCIPLOT or van der Waals contact mapping. Such analyses would offer deeper mechanistic insight into the stereochemical control exerted by the chiral catalyst.

Response: According to the suggestions, we have performed energy decomposition analysis (EDA) using ADF (2025.102) package at the B3LYP(D3BJ)/DZP level of theory. The EDA results summarized in Figure 1 clearly demonstrate that orbital interactions are the primary factor in favor of *TS1-SR* over *TS1-RS*. ($\Delta\Delta E_{\text{orb}} = -14.8$ kcal/mol).

The EDA-NOCV method involves the decomposition of the intrinsic interaction energy (ΔE_{int}) of transition states into electrostatic energy (ΔE_{elstat}), Pauli repulsion energy (ΔE_{Pauli}), orbital interaction energy (ΔE_{orb}) and dispersion energy (ΔE_{disp}). The energy components of four representative transition states (*TS1-SR*, *TS1-RS*, *TS1-RR* and *TS1-SS*) are summarized in Table 1. Among them, *TS1-SR* exhibits the most favorable interaction energy ($\Delta E_{\text{int}} = -90.5$ kcal/mol), primarily driven by strong orbital interactions ($\Delta E_{\text{orb}} = -108.0$ kcal/mol). Electrostatic interactions also contribute significantly to the stabilization ($\Delta E_{\text{elstat}} = -96.5$ kcal/mol). Although Pauli repulsion is inherently destabilizing ($\Delta E_{\text{Pauli}} = 158.6$ kcal/mol), its effect is effectively counterbalanced by the more stabilizing orbital and electrostatic components in this configuration. Dispersion interactions provide a moderate contribution (ranging from -42.4 to -44.6 kcal/mol) with minimal variation across the transition states. These results and discussions have been included in SI.

Figure 1. The EDA results for *TS1-SR* and *TS1-RS*.

Table 1. Energy decomposition analysis of four representative transition states. The unit of energy is kcal/mol.

	ΔE_{int}	ΔE_{elstat}	ΔE_{Pauli}	ΔE_{orb}	ΔE_{disp}
TS1-SR	-90.5	-96.5	158.6	-108.0	-44.6
TS1-RS	-84.5	-85.5	138.7	-93.2	-44.4
TS1-RR	-79.1	-77.3	125.2	-84.7	-42.4
TS1-SS	-86.6	-77.1	123.5	-88.7	-44.3

(2) *Remark 2:* Related to the above, it appears that the authors did not conduct a thorough exploration of the conformational space for the reactive intermediate (structure **13** in Fig. 5a) and the TS structures. This may have led to a bias in selecting conformations that best support the proposed mechanism. Consequently, the reported $\Delta\Delta G_{\ddagger}^{\ddagger}$ values may be overestimated due to the omission of low-lying competing TS structures. Therefore, the authors are strongly encouraged to perform a more comprehensive conformational search for all relevant species to ensure the reliability of the computed selectivity.

Response: According to the suggestion, we have conducted the comprehensive conformational search. The various conformers of **13** along with their relative Gibbs free energies are presented in Figure 2. Based on these results, **s13-1** was identified as the most stable form and has therefore used to in the following transition state calculations.

Figure 2. Conformational search and relative Gibbs free energies (kcal/mol) for structure **13**

Eight representative transition states and the relative Gibbs free energies are given in

Figure 3. *TS1-SR* (numbered **15** in the main text) was identified as the most stable one. These results and discussions have been included in SI.

Figure 3. Conformational search and relative Gibbs free energies (kcal/mol) for transition states.

(3) *Remark 3: A notable experimental observation is the exclusive formation of the α -C-H alkylation product, with no evidence of N-alkylation of the benzylamine via a direct $SN2'$ pathway. However, the authors do not address the observed chemoselectivity through DFT calculations. I recommend including a computational comparison between the imine-formation pathway and a plausible $SN2'$ attack of the free benzylamine on the MBH acetate-derived intermediate. Addressing this aspect is crucial to provide mechanistic insights that could guide related transformations.*

Response: According to the suggestion, we performed calculations for the following two pathways: α -C-alkylation and N-alkylation. As shown in Figure 4. The results indicate that α -C-Alkylation of benzylamine with MBH-adduct-derived intermediate **14** is energetically more favorable, which aligns well with the experimentally observed chemoselectivity. These results and discussions have been included in SI.

Figure 4. The potential energy surfaces.

(4) *Remark 4: The proposed mechanism states that, upon imine formation between the catalyst and the benzylamine, the acidic C–H bond at the benzylic position is selectively deprotonated to generate the reactive carbanion in situ. However, the authors do not provide a rationale to explain why this deprotonation is favored over alternative acidic sites, particularly the phenolic OH group of the pyridoxal scaffold. A computational pK_a comparison or an energy profile for the competitive deprotonation pathways would strengthen the mechanistic proposal.*

Response: According to the suggestion, we carried out a computational pK_a comparison, and the results are summarized in the following table. The estimated pK_a values for the benzylic C-H, phenolic O-H and N-H bonds in compound **12** are 17.7, 21.1 and 23.3, respectively. These results indicate that the benzylic C–H bond is the most acidic among the three sites. Therefore, deprotonation at this position is thermodynamically favored, providing a rationale for the selective formation of the reactive carbanion in our proposed mechanism. These results and discussions have been included in the main text and SI.

Reference	 27 pK _a (C-H) = 24.3	 29 pK _a (O-H) = 15.7	 31 pK _a (N-H) = 18.8
X-H	 12 pK _a (C-H) = 17.7	 12 pK _a (O-H) = 21.1	 12 pK _a (N-H) = 23.3

(5) *Remark 5: As a final remark, I strongly recommend reoptimizing the key stereodetermining TS structures using a range-separated hybrid functional such as ωB97X-D, which has shown excellent performance in capturing dispersion and electrostatic interactions relevant in organocatalysis. Moreover, I recommend that the Gibbs free energies discussed in the manuscript and Supporting Information be*

corrected using the quasi-rigid rotor harmonic oscillator (QRRHO) approach, as proposed by Grimme, to provide more accurate thermodynamic values.

Response: According to the suggestion, we carried out transition state calculations by using the ω B97X-D functional with the 6-311G(d,p) basis set. Gibbs free energy corrections were applied using the quasi-rigid rotor harmonic oscillator (QRRHO) approach proposed by Grimme. The thermal free energy corrections were performed using the Shermo software. The optimized three-dimensional geometries of four transition states, along with their calculated relative Gibbs free-energy values (kcal/mol), are presented in the following figure. The transition state TS3-SR displays the lowest Gibbs free energy, which is consistent with the calculation by using B3LYP(D3BJ) functional. These results and discussions have been included in SI.

POINT-TO-POINT RESPONSE:

Response to Reviewer #1:

(1) *The authors had addressed issues suggested by the reviewers. The revised manuscript looks fine. The reviewer recommends accepting the manuscript for publication in this journal.*

Response: Great thanks for the positive feedback and kind words.

Response to Reviewer #3: Great thanks for the helpful comments and suggestions

The authors have carefully and satisfactorily addressed all the concerns raised in the initial round of review. The revised version provides a significantly improved mechanistic description, supported by new DFT calculations. Therefore, I find the current version of the manuscript suitable for publication in Nature Communications. However, I would like to raise two minor points that the authors may consider prior to publication.

(1) *In response to Remark 3, the authors computed the transition state for the addition of benzylamine to the MBH adduct, reporting an activation free energy (ΔG^\ddagger) of 20.8 kcal/mol. While this information is valuable, the manuscript would benefit from a comparison with the barrier for the addition of benzylamine to the pyridoxal derivative (S,S)-6a, leading to intermediate 12, as previously suggested by this reviewer. Such a comparison would provide a more complete view of the chemoselectivity in the early stage of the mechanism.*

Response: According to the suggestions, we conducted a computational comparison of two pathways: (i) the addition of benzylamine to the pyridoxal catalyst (S,S)-6a to form imine 12, and (ii) the S_N2 attack of free benzylamine on intermediate 14. The corresponding potential energy surfaces are presented in Figure S1. The results reveal that the addition of benzylamine to (S,S)-6a, leading to intermediate 12, is energetically more favorable. This finding is consistent with the observed chemoselectivity. These results and discussions have been included in SI.

Figure S1. The potential energy surfaces.

Table S1. Enthalpies and Gibbs free energies (Hartree) of related structures in Figure S1.

Geometry	H	G
Cat	-1532.822124	-1532.900642
DBU	-462.063548	-462.099891
H ₂ O	-76.44531	-76.462975
TS4	-1859.741909	-1859.83265
TS5	-2321.834535	-2321.942692
TS6	-2321.821434	-2321.928238
TS7	-1783.302283	-1783.391709
Int1	-1859.739595	-1859.83097
Int2	-2321.838255	-2321.947461
Int3	-1783.301862	-1783.392578
Int4	-1783.305061	-1783.394432

(2) *Regarding the EDA analysis, the authors identify that the orbital interaction term (ΔE_{orb}) is significantly more stabilizing in TS1-SR compared to TS1-RS. At first sight, this difference could be ascribed to a shorter forming C–C bond in TS1-SR. In this context, it would be informative to explore whether noncovalent interactions or steric effects are responsible for limiting a more favorable nucleophilic approach in TS1-RS. A qualitative analysis using NCI-plot could help clarify whether the spatial arrangement of aryl groups in TS1-RS hinders optimal orbital overlap through repulsive interactions.*

Response: According to the suggestion, we have conducted a qualitative analysis using NCI plots (Figure S2a). NCI-plots revealed the presence of van der Waals interactions in both **TS1-SR** and **TS1-RS** qualitatively. No significant steric repulsion was observed between the aryl substituents and nearby functional groups in **TS1-RS**. A structural comparison of **TS1-SR** and **TS1-RS** is shown in Figure S2b. The H-bond length of NH \cdots O=C is 1.87 Å in **TS1-SR**, and it is 1.94 Å in **TS1-RS**. This shorter hydrogen bond in **TS1-SR** indicates that the hydrogen bond in **TS1-SR** is stronger than in **TS1-RS**, which partly contributes to the stronger orbital interaction in **TS1-SR**. In addition, the stronger hydrogen bond in **TS1-SR** shortens the distance between the imine anion and the MBH acetate intermediate, leading to better orbital overlap and enhanced orbital interaction. These results and discussions have been included in SI.

Figure S2. (a) NCI-plots of TS1-SR and TS1-RS; (b) Geometries of TS1-SR and TS1-RS.